# Efficacy and cost effectiveness of intravenous ferric carboxymaltose versus iron sucrose in adult patients with iron deficiency anaemia

Ahmad Basha[1]*, Mohamed Izham Mohamed Ibrahim[2], Anas Hamad[1], Prem Chandra[1], Nabil E. Omar[1], Mohamed Abdul Jaber Abdullah[3], Mahmood B. Aldapt[3], Radwa M. Hussein[1], Ahmed Mahfouz[4], Ahmad A. Adel[1], Hawraa M. Shwaylia[3], Yaslem Ekeibed[3], Rami AbuMousa[1], Mohamed A. Yassin[3]

1 National Center for Cancer Care and Research (NCCCR), Pharmacy Department, HMC, Doha, Qatar, 2 Clinical Pharmacy and Practice Department, College of Pharmacy, QU Health, Qatar University, Doha, Qatar, 3 National Center for Cancer Care and Research, Hematology Department, HMC, Doha, Qatar, 4 Heart Hospital (HH), Pharmacy Department, HMC, Doha, Qatar

* bashaahmed2222@yahoo.com

**Data Availability Statement:** All relevant data are within the manuscript and its Supporting information files.

## Abstract

### Background

Iron deficiency anaemia (IDA) is a major health issues and common type of nutritional deficiency worldwide. For IDA treatment, intravenous (IV) iron is a useful therapy.

### Objective

To determine the efficacy and cost-effectiveness (CE) of intravenous (IV) Ferric Carboxymaltose (FCM) versus IV Iron Sucrose (IS) in treating IDA.

### Data sources

Electronic medical record i.e. Cerner® system.

### Target population

Adults patients with iron deficiency anaemia.

### Time horizon

A 12-month period (01/01/2018–31/12/2018).

### Perspective

Hamad Medical Corporation (HMC, a public hospital).

### Intervention

IV Ferric Carboxymaltose versus IV Iron Sucrose.

**Funding:** The author(s) received no specific funding for this work.

**Competing interests:** The authors have declared that no competing interests exist.

## Outcome measures

With regard to responses to treatment i.e., efficacy of treatment with FCM & IS in IDA patients, hemoglobin (Hgb), ferritin, and transferrin saturation (TSAT) levels were the primary outcomes. Additionally, the researchers also collected levels of iron, platelet, white blood cell (WBC), red blood cell (RBC), mean corpuscular hemoglobin (MCH), and mean corpuscular volume (MCV). The costs i.e. resources consumed (obtained from NCCCR-HMC) and the CE of FCM versus IS were the secondary outcomes.

## Results of base-case analysis

There was a significant improvement in Hgb, RBC and MCH levels in the IS group than the FCM group. The overall cost of IS therapy was significantly higher than FCM. The medication cost for FCM was approximately 6.5 times higher than IS, nonetheless, it is cheaper in terms of bed cost and nursing cost. The cost effectiveness (CE) ratio illustrated that FCM and IS were significantly different in terms of Hgb, ferritin and MCH levels. Further, Incremental Cost Effectiveness Ratio (ICER) indicated that further justifications and decisions need to be made for FCM when using Hgb, iron, TSAT, MCH and MCV levels as surrogate outcomes.

## Results of sensitivity analysis

Not applicable.

## Limitations

The study did not consider the clinical or humanistic outcome.

## Conclusions

The higher cost of FCM versus IS can be offset by savings in healthcare personnel time and bed space. ICER indicated that further justifications and decisions need to be made for FCM when using Hgb, iron, TSAT, MCH and MCV levels as surrogate outcomes.

## Introduction

Iron is the most common nutritional deficiency worldwide, especially in developing countries which can lead to anaemia [1]. The WHO reports have demonstrated that anaemia affects 1.59 billion people, which constitutes to 24% of the total population [2]. Anaemia is defined by the WHO as a reduction in hemoglobin less than 13 g/dL (130 g/L) in men older than age 15 years, <12 g/dL (120 g/L) in non-pregnant women older than age 15 years, and <11 g/dL (110 g/L) in pregnant women [3]. Iron deficiency anaemia (IDA) represents the most prevailing kind of anaemia worldwide, and it is accompanied by depleted iron stores and signs of a compromised supply of iron to the tissues which may include fatigue, pale skin, chest pain, headache, and dizziness [4].

The options for managing IDA are explained by Short et al. (2013) [5]. For treatment, IV iron is an effective therapy. Furthermore, oral iron supplementation indicated quite promising results in increasing hemoglobin levels, nevertheless, gastrointestinal side effects like nausea, constipation, gastritis are likely to affect compliance in a high percentage of patients [6].

Intravenous iron preparations have been used for treating IDA, with promising result and avoiding blood transfusion as well as side effects of the oral iron preparation. Iron sucrose (IS) has been widely used for treating IDA showing a potential result. The efficacy and safety of IS have been demonstrated, as It has good bioavailability and a low incidence of anaphylaxis (Johnson-Wimbley, 2011) [7]. However, multiple doses are required for IS which decreases compliance. Ferric carboxymaltose (FCM) is non-dextran containing intravenous iron agent, having low immunogenic potential, and thus is not predisposed to a high risk of anaphylactic reactions [8], designed to be administered in large doses in a short period, with less side effects while overcoming the limitations of the existing intravenous iron agents. The safety and efficacy of FCM in patients with IDA have been reported with significant improvement in Hgb level [9]. The acquisition cost of FCM is expensive; however, better results have been reported compared to IS and other oral iron supplements. FCM has been reported to be cost effective and requires less frequent hospital visits that improves patient compliance [10–12].

In Qatar, Hamad Medical Corporation (HMC), is the main public healthcare institutions that serve about 3 million of the country's population. HMC provides different preparations of iron supplements that are used for the treatment of iron-deficiency anaemia, varying between oral and IV formulas, in addition to blood transfusion which is an option for more severe cases of anaemia. Both IS and FCM require special care and close monitoring in administration, hence all of the recruited patients purchased parenteral iron supplement from any pharmacy in HMC, and then went to the IV suite in HMC medical city where the injection and administration process was done. All patients are eligible for both medications. With regard to the Qataris and patients from the Gulf countries, medications are free, whereas residents have to pay 20% of the medication cost unless they have HMC insurance, then they have to pay 10%, while the non-residents have to pay 100% of the cost.

To our knowledge, studies that evaluated the efficacy and CE of IV FCM versus IS in adult patients with IDA are limited. Both medications need special care and close monitoring during administration. Consequently, this study was designed to test the efficacy and CE of FCM versus IS in treating IDA.

## Objectives

The current study aimed to:

1. Determine the efficacy of FCM in comparison to IS injection in treating iron deficiency anaemia (IDA) using laboratory assays (e.g., Hgb, ferritin and TSAT levels) as measuring parameters between the two groups.

2. Compare and evaluate CE for IV FCM and IS in the treatment of IDA patients.

## Methods

### Study design

This was a cross-sectional study with retrospective data. The study reviewed the efficacy of IV FCM versus IV IS in adult patients with IDA and performed CE evaluation as well as ICER. The patients were followed for a 12-month period. The therapeutic regimen used of FCM is between 25–100 mg IV daily PRN to a maximum cumulative dose of 1500 mg of iron, or 1–2 tablets of IS per day (to achieve 100–200 mg elemental iron daily in divided doses). The therapeutic treatment dose for each patient depends on different criteria like weight and HB level. Clinical, laboratory and cost data were collected before and after the first drug injection. The

study was carried out at hospitals under HMC. Moreover, the study perspective was HMC, which is a national healthcare institution.

## Study population and sample size

The targeted anaemic population was adult patients with IDA who were followed up with the medical city IV suite between 01/01/2018–31/12/2018. Patients were diagnosed as having IDA if the Hb <10–11.5 g/dl for women and <12.5–13.8 g/dl for men. Patients who were 18 years of age or older with IDA were included in the current study, besides they received FCM or IS for treatment of IDA. With the patient record system–Cerner[®], we excluded patients if they are less than 18 years old. Pregnant women, prisoners, and patients with anaemia rather than IDA, as well as participants with a known history of allergy to injection iron, were also excluded from the present study. The database system allowed the researchers to filter and sort patients in order to exclude other patients with different types of anaemia. This is a retrospective descriptive study that attempted to include all patients who met the inclusion criteria during the study period due to the lack of similar studies and accurate findings in literature; however, it was not possible to compute a formal sample size for the current research. From a previous earlier study by Christoph et al. (2012), the final Hgb after administering iron sucrose Hgb increased from 95.6 g/L to 110.4 g/L with a standard deviation of 11.9, taking into account the differences in the mean of non-inferiority limit of mean of Hgb between the two groups as 10 g/L, and the expected mean difference as zero, and a standard deviation as 11.9, for each group sample of 98 is required (i.e. a total sample size of 196, assuming equal group sizes) [13]. The findings of the current research reported a power of 90% and a level of significance of 5%, consequently it was concluded that the FCM is not inferior to the IS group at -5 units margin of non-inferiority (assuming that a larger mean is desirable), considering a 10% loss to follow-up, therefore at least 107 iron-deficient anemic adults need to be included in each study group.

## Outcome measures

With respect to responses to treatment i.e., the efficacy of treatment with FCM & IS in IDA patients, Hgb, ferritin, and TSAT levels were the primary outcomes. We aimed to restore these levels to normal (as demonstrated in Table 2). The Ganzoni formula was used by the lab to calculate iron deficiency. Hemoglobin level check was done at hematology lab. Ferritin was measured by an electrochemiluminescence immunoassay "ECLIA" done on the ROCHE cobas e 801 immunoassay analyzer based on sandwich principle. Iron saturation is a calculated parameter based on iron and transferrin (i.e. Iron Saturation % = Iron ÷ (Transferrin x 25.1) x 100). Additionally, the researchers also collected the levels of iron, platelet, WBC, RBC, MCH and MCV. The costs i.e., resources consumed (obtained from NCCCR-HMC) and the CE of FCM versus IS, were the secondary outcome. The cost-effectiveness ratio is defined as: [Total cost of resources consumed / % difference of changes or improvement from baseline]. While, ICER was estimated as below: [14, 15].

$$[\Delta C \ = \ C_I - \ C_0] \ / \ [\Delta \ = \ E_I - \ E_0]$$

Where, $C_I$ = cost of iron carboxymaltose.
$C_0$ = cost of iron saccharate.
$E_I$ = efficacy of iron carboxymaltose.
$E_0$ = efficacy of iron saccharate.
($\Delta C = C_I - C_0$ also referred to as the incremental cost) and ($\Delta E = E_I - E_0$ also referred to as the incremental effect). If the incremental cost is negative and the incremental effect is positive, then the intervention is unequivocally cost-effective (it is dominant, achieving better outcomes

at lower cost). In contrast, if the incremental cost is positive and the incremental effect is negative, then the intervention is not unequivocally feasible (it is dominated, achieving poorer outcomes at higher cost) [16]. In such case, the option should be rejected.

## Data collection method

The duration of the study was one year after obtaining the ethical approval. Data were collected from 01/06/2017 to 01/04/2019 (approximately 23 months). All the recruited patients purchased the parenteral iron supplement from any pharmacy in HMC (NCCCR, Al Amal Hospital) pharmacy, women's hospital pharmacy, Hamad General hospital pharmacy, etc.. . ..), and then went to IV suite in HMC medical city where the injecting and administration process were done. The primary data and records for, instance; health card number, gender, age were collected from IV suite log book. The pre-clinical and post-readings of the variables, were collected from the electronic health record system (Cerner®) after the approval by The Medical Research Center (MRC). The pre-level was measured close to medication administration, whereas the post-level was measured at least two weeks after medication administration. The study compared response to therapy based on the improvement in Hgb, ferritin and TSAT levels to evaluate the effectiveness of FCM and IS in treating IDA patients 6 months before and 4 months after the two therapies. The costs data were obtained from the HMC payer cashier system.

In general, the most common side effect is hypersensitivity or anaphylactic reactions, thus patients were monitored for more than 30 minutes following the end of administration and until clinically stable. Emergency equipment and medications (e.g., diphenhydramine injection, dexamethasone injection, etc. . .) were at the bedside in case a serious reaction. The patient also may experience nausea, dizziness, vision changes, headache, injection site skin discoloration and transient hypertension during and following the infusion for 30 minutes. All these observations are noted.

## Data analysis

Data were captured from the CERNER® system and extracted in Excel. All statistical analyses were carried out using the statistical packages SPSS 26.0 (IBM SPSS Statistics for Windows, Version 26.0. Armonk, NY: IBM Corp.). Descriptive statistics summarized and determined the sample demographic, clinical, laboratory including CBC profiles and other related features of patients. Normally distributed data were reported with mean and standard deviation (SD) with corresponding 95% confidence interval (CI), whereas the remaining results were reported with median and interquartile range (IQR). Categorical data were summarized using frequencies and percentages. Associations between two or more qualitative variables were examined and assessed using Pearson Chi- square and Fisher Exact tests. Efficacy measures included change in Hgb, ferritin, and TSAT from the baseline and were calculated using Wilcoxon signed rank test. Two-Way Repeated Measures ANOVA (Mixed-Factor ANOVA) was applied to compare between before and after effect in the two treatment arms. The Mann Whitney U test and Kruskal-Wallis test were used to compare the means of study groups. Cost effectiveness and ICER were analyzed using the aforementioned statistical methods. A two-sided P value <0.05 was considered to be statistically significant.

## Ethical consideration

The study was approved by HMC-IRB (i.e. MRC approval number 2101783; Protocol Number: MRC-01-19-171. The patient records and all data used in our retrospective study were fully anonymized before we transferred to SPSS program for data analysis and reporting.

**Table 1. Demographic profiles of patients in the two treatment arms.**

| Characteristics | | Ferric carboxymaltose (n = 396) | Iron sucrose (n = 368) | p value |
|---|---|---|---|---|
| Age | Mean ± SD | 41.6 ± 12.8 | 41.3 ± 12.4 | 0.716* |
| Gender | Female, n (%) | 368 (92.9) | 347 (94.3) | 0.442** |
| | Male, n (%) | 28 (7.1) | 21 (5.7) | |

Note:

* = independent t-test;

** = chi-square test

## Results

### (i) Demographic profile of patients

Table 1 below demonstrates the basic demographic profiles of the patients in the two treatment arms (764 patients). Mean age was similar and not significantly different (p = 0.716). There was a greater number of female patients in both groups and a non-significant difference in terms of gender (p = 0.442).

### (ii) Clinical and adverse outcomes of patients

The clinical characteristics and treatment outcomes are illustrated in Table 2. Comparisons of high significant are number of injections (p = 0.0001), number of ampoules received (p = 0.0001), number of normal saline 0.9% 100 mL bags with IV set (p = 0.0001) and number of visits to IV suite (p = 0.0001). In general, these numbers were higher in the IS group. When we used two-Way ANOVA for repeated measures to analyze levels of laboratory data before and after treatment, significant differences were detected for Hgb level (p = 0.0001), TSAT level (p = 0.045), platelet count (p = 0.001), and MCH (p = 0.0001) and MCV (p = 0.0001). Hypophosphatemia is the most common adverse reaction that occurred with IV iron supplement; however, it was not as serious as anaphylactic reactions. The study focuses only on hypersensitivity as customary done in our practice.

### (iii) Changes in clinical outcomes

Table 3 below shows the changes/differences in clinical outcomes/laboratory data for the two groups after treatment. When we compared these two groups using Mann-Whitney test, there were statistically significant differences detected only by hemoglobin levels (p = 0.0001), RBC (p = 0.001) and MCH (p = 0.0001). For both data, changes in the IS group were higher than in the FCM group.

### (iv) Resources consumed comparison

The cost analysis is shown in Table 4. The total cost of FCM was slightly lower than IS and significant (p = 0.0001). The medication cost of FCM was higher (around 7 times) than IS and significant (p = 0.0001); however, the bed cost was significantly higher (approximately 4 times) for IS than FCM (p = 0.0001). Furthermore, the analysis showed that the nursing cost for IS was nearly 4 times higher than FCM (p = 0.0001).

**Table 2. Clinical characteristics and treatment outcomes for patients in the two treatment arms.**

| Characteristics | | Ferric carboxymaltose (n = 396) | Iron sucrose (n = 368) | p value |
|---|---|---|---|---|
| Height | Mean ± SD | 159.8 ± 7.8 | 158.4 ± 7.1 | 0.076* |
| | Median (IQR) | 159.0 (9.0) | 158.0 (8.0) | |
| Weight | Mean ± SD | 78.0 ± 18.6 | 78.4 ± 18.8 | 0.440* |
| | Median (IQR) | 76.0 (23.6) | 76.0 (22.0) | |
| Number of injections | Mean ± SD | 1.8 ± 0.6 | 8.5 ± 4.4 | **0.0001*** |
| | Median (IQR) | 2.0 (1.0) | 8.0 (5.0) | |
| Number of ampoules received | Mean ± SD | 1.8 ± 0.6 | 8.5 ± 4.4 | **0.0001*** |
| | Median (IQR) | 2.0 (1.0) | 8.0 (5.0) | |
| Number of NS 0.9% 100ml bags with IV set | Mean ± SD | 1.3 ± 0.5 | 5.7 ± 2.8 | **0.0001*** |
| | Median (IQR) | 1.0 (1.0) | 5.0 (5.0) | |
| Number of visit to IV suite (BED COST) | Mean ± SD | 1.3 ± 0.5 | 5.7 ± 2.8 | **0.0001*** |
| | Median (IQR) | 1.0 (1.0) | 5.0 (1.0) | |
| Hgb (g/dL) | before | 9.8 ± 1.7 | 9.0 ± 1.7 | **0.0001**** |
| | after | 11.8 ± 1.5 | 11.8 ± 1.3 | |
| Ferritin (mg/L) | before | 30.4 ± 73.8 | 25.2 ± 86.1 | 0.579** |
| | after | 232.0 ± 375.4 | 218.8 ± 262.8 | |
| Iron (μmol/L) | before | 6.6 ± 6.2 | 5.2 ± 5.4 | 0.916** |
| | after | 13.2 ± 12.4 | 13.5 ± 8.3 | |
| Transferrin (mg/dL) | before | 3.4 ± 6.1 | 3.4 ± 4.2 | 0.156** |
| | after | 2.4 ± 0.6 | 2.8 ± 4.1 | |
| Transferrin saturation (%) | before | 10.7 ± 10.9 | 7.2 ± 5.6 | **0.045**** |
| | after | 25.8 ± 17.0 | 22.8 ± 11.5 | |
| Platelet (per L) | before | 304.7 ± 102.4 | 334.2 ± 109.8 | **0.001**** |
| | after | 267.0 ± 84.1 | 284.5 ± 83.7 | |
| WBC (per L) | before | 6.4 ± 2.2 | 6.9 ± 2.4 | 0.05** |
| | after | 6.5 ± 2.6 | 6.7 ± 2.2 | |
| RBC (cells/mcL) | before | 4.3 ± 0.6 | 4.2 ± 0.6 | 0.818** |
| | after | 4.6 ± 0.6 | 4.7 ± 1.1 | |
| MCH (pg) | before | 30.4 ± 18.2 | 21.5 ± 4.0 | **0.0001**** |
| | after | 35.5 ± 22.6 | 26.2 ± 3.9 | |
| MCV (fL) | before | 65.4 ± 21.1 | 71.4 ± 9.8 | **0.0001**** |
| | after | 73.1 ± 21.7 | 81.0 ± 7.8 | |

Note:

* Mann-Whitney test;

** Two-Way Repeated Measures ANOVA (Mixed-Factor ANOVA); p values in bold mean significant

Normal values:

Hemoglobin: 13–17 g/dL (men), 12–15 g/dL (women); Ferritin: 12–300 ng/mL (men), 12–150 ng/mL (women); Iron: 10.74 to 30.43 micromoles per liter (micromol/L); Transferrin: 200–350 mg/dL; Transferrin saturation (%): 20%–50%; Platelets: 150–400 x 10^9/L; White blood cells (WBC) 4–10 x 10^9/L; RBC: Male: 4.7 to 6.1 million cells per microliter (cells/mcL) Female: 4.2 to 5.4 million cells/mcL; Mean corpuscular hemoglobin (MCH): 27.5 and 33.2 picograms (pg); Mean corpuscular volume (MCV): 80–100 fL

## (v) Cost-economic analysis

Table 5 illustrates the economic analysis we carried out in the two treatment arms. The CE analysis and ICER were carried out between FCM and IS for each of the clinical outcomes. We were unable to show significant differences of in vitro data between FCM and IS except for

**Table 3. Changes in the comparison of clinical outcomes between patients in the two treatment arms.**

| Changes/Difference in effectiveness (%) | | | Ferric carboxymaltose (n = 396) | Iron Sucrose (n = 368) | p value* |
|---|---|---|---|---|---|
| | | | Outcome | Outcome | |
| Hgb (g/dL) | (before-after) | Mean (95% CI) | 25.09 (20.80–29.37) | 29.03 (23.33–34.72) | **0.0001** |
| | | Median (IQR) | 19.19 (28.29) | 20.79 (25.39) | |
| Ferritin (mg/L) | (before-after) | Mean (95% CI) | 3328.85 (1200.96–5456.75) | 1659.25 (972.36–2346.14) | 0.115 |
| | | Median (IQR) | 814.98 (2767.70) | 666.67 (1888.67) | |
| Iron (μmol/L) | (before-after) | Mean (95% CI) | 181.96 (130.53–233.40) | 260.45 (154.15–366.74) | 0.111 |
| | | Median (IQR) | 183.33 (235.71) | 140.58 213.45) | |
| Transferrin (mg/dL) | (before-after) | Mean (95% CI) | -16.72 (-20.82-(-12.62)) | -17.94 (-23.57-(-12.30)) | 0.868 |
| | | Median (IQR) | -16.00 (29.17) | -17.58 (21.40) | |
| Transferrin saturation (%) | (before-after) | Mean (95% CI) | 284.21 (177.01–383.45) | 327.15 (204.00–450.29) | 0.079 |
| | | Median (IQR) | 140.0 (316.14) | 160.00 (341.79) | |
| Platelet (per L) | (before-after) | Mean (95% CI) | -9.47 (-12.10-(-6.85)) | -11.19 (-13.69-(-8.68)) | 0.236 |
| | | Median (IQR) | -12.14 (22.10) | -13.33 (24.28) | |
| WBC (per L) | (before-after) | Mean (95% CI) | 5.74 (1.33–10.15) | 1.75 (-1.73–5.52) | 0.481 |
| | | Median (IQR) | -2.64 (30.34) | -3.07 (33.21) | |
| RBC (cells/mcL) | (before-after) | Mean (95% CI) | 5.64 (4.51–6.78) | 10.92 (7.92–13.91) | **0.001** |
| | | Median (IQR) | 4.82 (11.61) | 6.74 (16.3) | |
| MCH (pg) | (before-after) | Mean (95% CI) | 18.71 (1198–25.45) | 23.19 (20.82–25.56) | **0.0001** |
| | | Median (IQR) | 12.08 (19.66) | 16.95 (30.25) | |
| MCV (fL) | (before-after) | Mean (95% CI) | 14.35 (12.33–16.38) | 14.60 (12.95–16.26) | 0.187 |
| | | Median (IQR) | 11.50 (15.41) | 11.15 (19.39) | |

Note: % difference or change = [(after treatment–before treatment)/before treatment] x 100;

Hgb (p = 0.001), ferritin (p = 0.042) and MCH (p = 0.001). With regard to ICER, it was negative for ferritin, transferrin, platelet, and WBC level. Whereas FCM was unequivocally ineffective (it was dominated, achieving poorer outcomes at higher cost). On the contrary, ICER was positive (i.e., the total cost of managing IDA patients with FCM is less expensive, nonetheless, it showed less improvement than IS) for Hgb, iron, TSAT, RBC, MCH and MCV levels. Thus, the decision has to be made with regard to these conditions i.e., the additional cost per extra unit of health effect for FCM, and overall, which makes it more expensive therapy.

## Discussion

The study aimed to analyze the efficacy as well as CE of IV FCM and IV IS injection in treating IDA. Our observations indicated that patients in the IS group used significantly higher number of injections, ampoules of medication, normal saline and visits to the IV suite compared to the FCM group. There were also significant changes in laboratory tests between the FCM and IS groups i.e., Hgb, TSAT, platelet count, MCH and MCV levels. Further analysis regarding the change in efficacy due to treatment, indicated that the changes of Hgb, RBC and MCH levels in the IS group were significantly higher than the FCM group. In terms of overall cost, IS was significantly higher than FCM, even though FCM is far more expensive than IS. The medication cost for FCM was approximately 6.5 times higher than IS; however, it was cheaper in terms of bed cost (approximately 4 times) as well as the nursing cost (approximately 5 times). The CE ratio illustrated that FCM and IS were significantly different in terms of Hgb, ferritin and MCH levels. Further, ICER indicated that further justifications and decisions need to be

**Table 4. Comparison of resource expenditures between patients in the two treatment arms.**

| Resources consumed (QAR) | | Ferric carboxymaltose (n = 396) | Iron sucrose (n = 368) | p value* |
|---|---|---|---|---|
| Medication cost | Mean ± SD | 720.20 ± 228.80 | 110.52 ± 56.81 | **0.0001** |
| | 95% CI | (697.60–742.81) | (104.69–116.35) | |
| | Median (IQR) | 800.00 (400.00) | 104.00 (65.00) | |
| Bed cost | Mean ± SD | 90.86 ± 34.30 | 399.40 ±194.66 | **0.0001** |
| | 95% CI | (87.47–94.25) | (379.42–419.38) | |
| | Median (IQR) | 70.00 (70.00) | 350.00 (350.00) | |
| Nursing cost | Mean ± SD | 104.43 ± 33.18 | 492.53 ± 253.34 | **0.0001** |
| | 95% CI | (101.15–107.71) | (466.58–518.50) | |
| | Median (IQR) | 116.00 (58.00) | 464.00 (290.00) | |
| Total cost | Mean ± SD | 915.49 ± 280.18 | 1002.99 ± 473.03 | **0.0001** |
| | 95% CI | (887.58–943.17) | (954.44–1051.55) | |
| | Median (IQR) | 986.00 (528.00) | 989.00 (633.00) | |

Note:

* Mann-Whitney test; figures in bold are significant at p < 0.001; USD 1 = QAR 3.65

**Table 5. Cost-economic analysis comparison between patients in the two treatment arms.**

| Changes/Difference in effectiveness | | | Ferric Carboxymaltose (n = 396) | Iron Sucrose (n = 368) | p value* | ICER (QAR) |
|---|---|---|---|---|---|---|
| | | | Cost/Effectiveness ratio | Cost/Effectiveness ratio | | |
| Hgb (g/dL) | (Before-after) | Mean (95% CI) | 45.34 (30.90–59.77) | 37.88 (21.34–54.43) | **0.001** | 22 |
| | | Median (IQR) | 38.29 (51.14) | 29.74 (39.29) | | Evaluate |
| Ferritin (mg/L) | (Before-after) | Mean (95% CI) | 2.51 (-0.36–5.38) | 0.14 (-1.42–1.70) | **0.042** | - 0 |
| | | Median (IQR) | 0.82 (2.93) | 0.55 (1.39) | | Accept |
| Iron (µmol/L) | (Before-after) | Mean (95% CI) | 13.27 (0.63–25.90) | 5.73 (0.26–11.19) | 0.474 | 1 |
| | | Median (IQR) | 4.31 (17.42) | 5.38 (9.87) | | Evaluate |
| Transferrin (mg/dL) | (Before-after) | Mean (95% CI) | -27.46 (-91.07–36.15) | -49.34 (-65.03 –(-33.66)) | 0.801 | - 72 |
| | | Median (IQR) | -35.20 (31.68) | -38.16 (49.43) | | Accept |
| Transferrin saturation (%) | (Before-after) | Mean (95% CI) | 8.33 (3.27–13.39) | 8.50 (3.40–13.61) | 0.360 | 2 |
| | | Median (IQR) | 4.49 (11.46) | 3.77 (6.70) | | Evaluate |
| Platelet (per L) | (Before-after) | Mean (95% CI) | -65.04 (-108.26 –(-21.81)) | -24.03 (-62.26–14.21) | 0.159 | -51 |
| | | Median (IQR) | -42.72 (75.19) | -37.40 (61.55) | | Accept |
| WBC (per L) | (Before-after) | Mean (95% CI) | -11.27 (-32.74–10.20) | -13.95 (-3772–9.83) | 0.910 | -22 |
| | | Median (IQR) | -23.71 (113.98) | -16.59 (100.54) | | Accept |
| RBC (cells/mcL) | (Before-after) | Mean (95% CI) | 77.92 (56.51–99.32) | 80.39 (55.62–105.15) | 0.486 | 17 |
| | | Median (IQR) | 79.20 (143.45) | 66.09 (119.43) | | Evaluate |
| MCH (pg) | (Before-after) | Mean (95% CI) | 91.70 (35.50–147.89) | 52.69 (4.66–100.73) | **0.001** | 20 |
| | | Median (IQR) | 59.41 (96.82) | 43.70 (74.21) | | Evaluate |
| MCV (fL) | (Before-after) | Mean (95% CI) | 189.43 (91.09–287.76) | 99.24 (70.27–128.21) | 0.926 | 350 |
| | | Median (IQR) | 67.39 (85.26) | 61.46 (122.90) | | Evaluate |

Note: % difference or changes = [(after treatment–before treatment)/before treatment] x 100;

* Mann-Whitney test; figures in bold are significant at p < 0.001

made for FCM when using Hgb, iron, TSAT, MCH and MCV levels. FCM was more expensive and the total cost of managing IDA was slightly less than IS, nevertheless, it was less effective than IS (i.e., SW quadrant in the ICER matrix). Thus, we need to decide if FCM is an efficient use of resources. Concerned healthcare providers and policymakers need to decide on the threshold value i.e., a maximum amount society is willing to spend for an incremental health improvement.

According to Jose et al. (2019), IDA is a major health issue and a common type of nutritional deficiency worldwide [17]. The 2015 WHO report indicated that the level of public health significance for anaemia among non-pregnant women and all women of reproductive age in Qatar is moderate [18]. Al Obaidely et al. (2017) reported that the prevalence of anaemia among elderly population in Qatar is high, with 60.3% classified as moderately severe [19]. IDA is a condition that can be avoided and treated (Zainel et al., 2018) [20]. Dillon et al. (2012) demonstrated that IV iron is a useful therapy for IDA treatment, [21]. Additionally, patients who are unresponsive or intolerant to oral iron can also benefit from this medication. IV iron preparations have been used for treating IDA, with promising result and making it possible to avoid blood transfusion and side effects of the oral iron preparation. Iron sucrose (IS) has been widely used for treating IDA as it showed potential result [22]. In our study, FCM and IS were compared in terms of efficacy by analyzing the surrogate (lab data changes) and economic outcomes. Our observations regarding the medication cost, safety and effectiveness of FCM versus IS, were similar as reported by Dillon et al. (2012) [21]. FCM is seven times more expensive than IS, besides no side effects were reported. Dillon et al. (2012) and Lee et al. (2019) found out that the increase in Hgb level in the FCM group was equivalent to patients treated with IS [21, 23]; however, this finding is inconsistent with our finding as IS showed better Hgb improvement. Lyseng-Williamson et al. (2009) reported that results of several randomized trials showed that intravenously administered FCM, rapidly improves Hgb levels and replenishes depleted iron stores in various populations of patients with IDA, including those with inflammatory bowel disease, heavy uterine bleeding, postpartum IDA or chronic kidney disease [24], in addition, it was well tolerated [24]. One of the advantages of FCM is convenient dosing with fewer total doses, which consequently will lead to better patient compliance [17]. Further, intravenous iron preparations have been used for treating IDA, with a promising result and making it possible to avoid blood transfusion and side effects of the oral iron preparation [25, 26]. However, treatment with oral iron is simple, cheaper and relatively effective. Friedrisch and Cancado (2015) further reported the disadvantages of the oral iron therapy e.g. GI adverse effects, limited absorption, lack of adherence due to the frequency of therapy and insufficient duration of therapy. On the other hand, they also mentioned the advantages of FCM such as better improvement of Hb, ferritin and transferrin saturation values, as well as patient's quality of life.

The economic analysis indicated controversial findings. As reported in other studies (21, 27), the higher cost of FCM can be offset by savings in healthcare personnel time and bed space which were similar with our previously mentioned results. Fragoulakis et al. (2012) conducted a cost-minimization analysis between FCM, IS and iron dextran in the treatment of IDA in Greece [27], and they found that the total cost of FCM was lower compared to the comparators. Therefore, FCM is suggested as a cost-saving option. FCM is shown to be non-inferior with similar or superior efficacy compared with oral iron.

The underlying cause of IDA should be addressed and iron therapy may be introduced to replenish iron stores. The data are supportive in terms of cost-effective treatment for patients not only to avoid the reduction in capacity for work due to anaemia, but also provide evidence of a cost-saving option for treating adults with IDA. IDA can cause problems such as physical and mental quality of life. Therefore, future studies with a larger sample size are needed, in

addition to investigating other health outcomes. More studies need to be carried out in order to compare these groups of medications and measure the clinical (e.g., alleviate the symptoms of iron deficiency) as well as humanistic (e.g. improve quality of life) outcomes.

There are a few limitations that fair to be considered. In general, the results of a CEA just offer an estimate of the extra cost for one additional patient outcome. The decision-maker's budget was not consider in economic assessments such as CEA. Thus, a decision-maker might deem a new treatment to be cost-effective but too expensive to accept. Calculations in health economics sometimes include value judgments that are not always clearly acknowledged. Moreover, the research did not specify the follow-up length (median or mean days of follow-up). This is relevant since the hemoglobin levels and iron parameters 2 weeks after the administration cannot be compared to values at a longer follow-up (e.g., 40 days). The follow-up between the two groups should be similar. Moreover, iron parameters at 2 weeks are still disrupted by the IV iron administration. Future prospective study should consider these points.

## Conclusion

In summary, treatment with FCM and IS indicated controversial findings. The higher cost of FCM versus IS can be overcome by savings in healthcare personnel time and bed space. The overall cost was slightly higher in the IS group. Hgb, TSAT, platelet count, MCH and MCV showed significant differences before and after treatment. Further findings indicated that the changes of Hgb and MCH levels in the IS group were significantly higher than in the FCM group. ICER indicated that further justifications and decisions need to be made for FCM when using Hgb, iron, TSAT, MCH and MCV levels as surrogate outcomes.

## Supporting information

**S1 Data.**
(XLSX)

## Acknowledgments

The authors express their gratitude to Prof. Ibrahim Al Janahi, Executive Director of Research, Hamad Medical Research Center. We would also like to thank the NCCCR-HMC management, administration and chronic clinic nurse team as well as physicians who supported this research.

## Author Contributions

**Data curation:** Nabil E. Omar, Mohamed Abdul Jaber Abdullah, Mahmood B. Aldapt, Radwa M. Hussein, Ahmed Mahfouz, Ahmad A. Adel, Hawraa M. Shwaylia, Yaslem Ekeibed, Rami AbuMousa.

**Methodology:** Prem Chandra.

**Supervision:** Anas Hamad, Mohamed A. Yassin.

**Writing – original draft:** Ahmad Basha, Mohamed Izham Mohamed Ibrahim.

**Writing – review & editing:** Ahmad Basha, Mohamed Izham Mohamed Ibrahim.

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
