## [Decision Letter · Decision Letter 0]

7 Jan 2021

PONE-D-20-35525

EFFICACY AND COST EFFECTIVENESS OF INTRAVENOUS FERRIC CARBOXYMALTOSE VERSUS IRON SUCROSE IN ADULT PATIENTS WITH IRON DEFICIENCY ANEMIA

PLOS ONE

Dear Dr. BASHA,

Thank you for submitting your manuscript to PLOS ONE. After careful consideration, we feel that it has merit but does not fully meet PLOS ONE’s publication criteria as it currently stands. Therefore, we invite you to submit a revised version of the manuscript that addresses the points raised during the review process.

As you can see, your manuscript received somewhat mixed reviews. Overall, I believe that it provides potentially useful data, but in its current form has multiple limitations. In particular, the selection of patients (inclusion/exlusion criteria), baseline characteristics, dosing strategy and many more need to be clearly described.

We look forward to receiving your revised manuscript.

Kind regards,

Pavel Strnad

Academic Editor

PLOS ONE

Journal Requirements:

3. In your ethics statement in the manuscript and in the online submission form, please provide additional information about the patient records used in your retrospective study.

Specifically, please ensure that you have discussed whether all data were fully anonymized before you accessed them.

4. Please ensure you have thoroughly discussed any potential limitations of this study within the Discussion section, including the potential impact of confounding factors.

6. We note that you have not completed the financial disclosure section with a funding statement.

7. We note that you have not completed the Competing Interests section of the submission form.

a. Please complete your Competing Interests statement to state any Competing Interests. If you have no competing interests, please state "The authors have declared that no competing interests exist.", as detailed online in our guide for authors at http://journals.plos.org/plosone/s/submit-now

Reviewers' comments:

Reviewer's Responses to Questions

**Comments to the Author**

1. Is the manuscript technically sound, and do the data support the conclusions?

Reviewer #1: No

Reviewer #2: Yes

Reviewer #3: Partly

2. Has the statistical analysis been performed appropriately and rigorously? 

Reviewer #1: I Don't Know

Reviewer #2: Yes

Reviewer #3: I Don't Know

3. Have the authors made all data underlying the findings in their manuscript fully available?

Reviewer #1: No

Reviewer #2: Yes

Reviewer #3: Yes

4. Is the manuscript presented in an intelligible fashion and written in standard English?

Reviewer #1: No

Reviewer #2: Yes

Reviewer #3: Yes

5. Review Comments to the Author

Reviewer #1: The study is certainly important for health care providers in Qatar but I believe that some major limitations in the study design warrant a work-over of the study.

Please specify outcome measures in the abstract. Improvement in laboratory levels is a meaningless statement.

The complete patient population is poorly defined. IDA is a disease spectrum. I recommend to clarify the patient selection (hemoglobin, ferritin concentration and transferrin saturation).

Was the iron deficit calculated using the Ganzoni formula? What kind of dosing strategy was used in the patient cohort? Was there a difference in total iron dose between groups?

Likewise, the exclusion criteria are poorly defended. How did the authors exclude other anemia causes? What kind of biochemical studies were routinely performed in the cohort?

Please provide a dedicated table of the baseline characteristics to show any differences between groups at baseline.

I have no experience in cost-effectiveness analyses and therefore cannot assess the statistical methods used.

Reviewer #2: Authors have here reported on the administration of two intra-venous iron formulations (ferric carboxy malthose and iron sucrose) in adult patients with iron deficiency anemia. This prospective study lasting 12 months is very interesting, well written and structured. Tables support the text. In detail this reviewer finds interesting the aspects related to RBC parameters improvement and cost (direct and indirect) analysis. O suggest to modify some aspects of the paper :

I think the variation in Hgb and other RBCs parameters for both treatments should be better highlighted by a specific figure where I suggest to report in detail only haemoglobin levels, MCV and ferritine.

I do not find useful reporting in table WBC and PLt counts, I do not see from data a particular high PLT count before treatment (reactive thrombocytosis) pre and post treatment values seem quite stable ,thus I think this should be just reported in the text. Further, it seems that patients treated with iron sucrose had baseline Hgb levels lower than patients treated with ferric carboxymalthose.

Could please authors better describe this aspect? Authors should alos better afford the issuesrelated to oral iron in comparison to iv iron.

References should be updated with a recently published article referring to the administration of iv iron vs oral iron(10.1007/s00277-020-04361-3)

Reviewer #3: In this study, the authors evaluate the efficacy and cost-effectiveness of ferric carboxymaltose compared to iron sucrose in adult patients with iron deficiency anemia.

Although clinically relevant due to the frequency of the disease and its impact, the work presents several major limitations:

- consider to reorganize the structure of the introduction (role of oral iron and intravenous iron) and include more References since many data are not supported by references (e.g., “Intravenous iron preparations have been used for treating IDA, with a promising result and making it possible to avoid blood transfusion and side effects of the oral iron preparation.” “Iron sucrose (IS) has been widely used for treating IDA that shows the potential result.”)

- The primary outcome, i.e., the treatment efficacy, is not well defined. Indeed, the authors consider hematological and iron parameters at a follow-up time point that was not clearly defined. “The post level was measured at least 2 weeks after the medication administration.” The authors do not specify the follow-up length (median or mean days of follow-up). This is relevant since the hemoglobin levels and iron parameters 2 weeks after the administration cannot be compared to values at a longer follow-up (e.g., 40 days). At least the follow-up between the two groups should be similar. Moreover, iron parameters at 2 weeks are still disrupted by the IV iron administration.

- The total amount of administered iron per patient in the two groups should be specified in order to compare the efficacy parameters.

- The two groups (FCM vs. IS) characteristics are not well defined: the total amount of iron administered per patient, causes of iron deficiency anemia, clinical symptoms. Moreover, their baseline hemoglobin levels are different. I suggest commenting on this difference and its potential impact on therapy choice. Nevertheless, since hemoglobin is the efficacy parameter, and hemoglobin increase is used for the cost-effectiveness evaluation, the authors should consider potential adjustments and comment on this.

- Adverse events data are not presented (even if not adverse events have been observed should be specified), while in the Methods section, there is a specific paragraph about the management of adverse reaction. Moreover, no data about hypophosphatemia have been presented. I would consider editing the part about drugs used to manage these reactions since the role of antihistaminic drugs is controversial.

Minor revisions:

- Introduction, page 4: the sentence “Anemia is the most common nutritional deficiency” should be revised. Indeed, iron deficiency is the most common nutritional deficiency.

- Introduction, page 4: “poor appetite” is not considered one of the most frequent symptoms of iron deficiency anemia

- Check all the abbreviations: some are spelled out many times, some are not spelled out

- “The acquisition cost of FCM is expensive but reported to have a better outcome.”: compared to?

- Units of measurements are not always specified in the tables

- Native English revision is required

6. PLOS authors have the option to publish the peer review history of their article (what does this mean?). If published, this will include your full peer review and any attached files.

Reviewer #1: No

Reviewer #2: No

Reviewer #3: No

---

## [Author Response · Author response to Decision Letter 0]

23 Apr 2021

all the required modifications and supporting files are covered and sent

1. our authors list is attached in the manuscript to ensure that each author is linked to an affiliation. i also modified the affiliation in the manuscript data section to be matched with the attached manuscript.

2. the data is available on the submission as a supporting file. 

the Supporting Information file contains the full data set needed to reach the conclusions drawn in the manuscript with related metadata and methods, and any additional data required to replicate the reported study findings in their entirety. This includes:

a) The values behind the means, standard deviations and other measures reported;

b) The values used to build graphs;

c) The points extracted from images for analysis.

---

## [Decision Letter · Decision Letter 1]

17 May 2021

PONE-D-20-35525R1

EFFICACY AND COST EFFECTIVENESS OF INTRAVENOUS FERRIC CARBOXYMALTOSE VERSUS IRON SUCROSE IN ADULT PATIENTS WITH IRON DEFICIENCY ANEMIA

PLOS ONE

Dear Dr. BASHA,

Thank you for submitting your manuscript to PLOS ONE. After careful consideration, we feel that it has merit but does not fully meet PLOS ONE’s publication criteria as it currently stands. Therefore, we invite you to submit a revised version of the manuscript that addresses the points raised during the review process.

As you can see, one of the reviewers has major concerns that have not yet been addressed and in fact recommended to reject the manuscript. Therefore, I strongly advise you to answer these comments as good as you can.

We look forward to receiving your revised manuscript.

Kind regards,

Pavel Strnad

Academic Editor

PLOS ONE

Reviewers' comments:

Reviewer's Responses to Questions

**Comments to the Author**

1. If the authors have adequately addressed your comments raised in a previous round of review and you feel that this manuscript is now acceptable for publication, you may indicate that here to bypass the “Comments to the Author” section, enter your conflict of interest statement in the “Confidential to Editor” section, and submit your "Accept" recommendation.

Reviewer #2: All comments have been addressed

Reviewer #3: (No Response)

2. Is the manuscript technically sound, and do the data support the conclusions?

Reviewer #2: Yes

Reviewer #3: Partly

3. Has the statistical analysis been performed appropriately and rigorously? 

Reviewer #2: Yes

Reviewer #3: I Don't Know

4. Have the authors made all data underlying the findings in their manuscript fully available?

Reviewer #2: Yes

Reviewer #3: No

5. Is the manuscript presented in an intelligible fashion and written in standard English?

Reviewer #2: Yes

Reviewer #3: Yes

6. Review Comments to the Author

Reviewer #2: I have revised this version of the manuscript, I think it has been improved , all comments have been well adrressed. I think the paper can now be accepted for pubblication

Reviewer #3: Considering the authors' comments at first revision, some crucial points have not been addressed:

- The author did not specify the follow-up lenght and this dramatically affect the comparison of efficacy parameters

- The authors did not specify the mean/median amount of iron (in milligrams) administered in the two groups and this dramatically affect the efficacy outcomes.

- The different Hb baseline level could affect therapeutic decisions

- The conclusion that IS showed a higher increase in Hb is affected by the Hb baseline difference between the two groups

7. PLOS authors have the option to publish the peer review history of their article (what does this mean?). If published, this will include your full peer review and any attached files.

Reviewer #2: No

Reviewer #3: No

---

## [Author Response · Author response to Decision Letter 1]

7 Jul 2021

June 30th, 2021

The Editor-In-Chief

PLOS ONE

Dear Editor and Reviewers,

Revision of Manuscript: Manuscript ID PONE-D-20-35525

EFFICACY AND COST EFFECTIVENESS OF INTRAVENOUS FERRIC CARBOXYMALTOSE VERSUS IRON SUCROSE IN ADULT PATIENTS WITH IRON DEFICIENCY ANEMIA 

The authors of the above manuscript would like to thank you for the email received which contained a few more comments for improvement. 

We are very pleased to know that it can be considered for publication contingent upon addressing the reviewers concerns and recommended edits. We really appreciate the time you spent reviewing this manuscript, and we thank you for all the comments. We have studied these comments carefully and have made corresponding corrections that we hope will meet with your approval. We believe once we addressed your comments, the quality of the manuscript will be improved.

Below kindly find our responses in regard to every comment mentioned by the reviewers.

At the end, we would like to thank the editor and the reviewers for considering this manuscript, and we look forward to hearing a favorable outcome from you soon. If you have any further queries, please do not hesitate to contact us.

Sincerely yours,

Dr Ahmed Basha

Corresponding author

Reviewers' comments:

Reviewer's Responses to Questions

Comments to the Author

1. If the authors have adequately addressed your comments raised in a previous round of review and you feel that this manuscript is now acceptable for publication, you may indicate that here to bypass the “Comments to the Author” section, enter your conflict of interest statement in the “Confidential to Editor” section, and submit your "Accept" recommendation.

Reviewer #2: All comments have been addressed

Reviewer #3: (No Response)

2. Is the manuscript technically sounds, and do the data support the conclusions?

Reviewer #2: Yes

Reviewer #3: Partly

3. Has the statistical analysis been performed appropriately and rigorously? 

Reviewer #2: Yes

Reviewer #3: I Don't Know

4. Have the authors made all data underlying the findings in their manuscript fully available?

Reviewer #2: Yes

Reviewer #3: No

Authors’ response: Dear Editor and Reviewer, The authors have submitted an excel sheet which contain all the data in our previous submission. Kindly, check and let us know if the sheet cannot be opened or found.

5. Is the manuscript presented in an intelligible fashion and written in standard English?

Reviewer #2: Yes

Reviewer #3: Yes

6. Review Comments to the Author

Reviewer #2: I have revised this version of the manuscript, I think it has been improved , all comments have been well addressed. I think the paper can now be accepted for publication

Reviewer #3: Considering the authors' comments at first revision, some crucial points have not been addressed:

- The author did not specify the follow-up length and this dramatically affect the comparison of efficacy parameters

- The authors did not specify the mean/median amount of iron (in milligrams) administered in the two groups and this dramatically affect the efficacy outcomes.

- The different Hb baseline level could affect therapeutic decisions

- The conclusion that IS showed a higher increase in Hb is affected by the Hb baseline difference between the two groups

- The author did not specify the follow-up length and this dramatically affect the comparison of efficacy parameters

Authors’ response: Dear reviewer, as stated in the method section, the authors have explained that the follow-up length was between June 2017 till Apr 2019 (around 23 months):

“A retrospective study was conducted to review the efficacy and cost effectiveness of intravenous (IV) ferric carboxymaltose (FCM) versus iron sucrose (IS) in adult patients with iron deficiency anemia (IDA) who are following with medical city IV suite between 01/01/2018 – 30/11/2018. They were included in the study and their labs including; hemoglobin, ferritin and TSAT were followed within the period from 01/06/2017 to 01/04/2019”

The study compared the response to the therapy using the improvement in hemoglobin, ferritin and TSAT levels to evaluate the effectiveness of FCM and IS in treating IDA patients, hemoglobin, ferritin and TSAT and other parameters that were obtained 6 months before and 4 months after the two therapies within the period from 01/06/2017 to 01/04/2019. 

Action: we will add ’23 months’ in brackets and the follow-up period in the data collection section.

- The authors did not specify the mean/median amount of iron (in milligrams) administered in the two groups and this dramatically affect the efficacy outcomes.

Authors’ response: The information was in the original ms submitted earlier. We have now highlighted for clarification. The information can be found in the study design section.

- The different Hb baseline level could affect therapeutic decisions

Authors’ response: Further, regarding the statistical analysis, we have ran analysis to control for the data at baseline and control for any differences. In the statistical analysis section, we wrote:

Efficacy measures included change in Hgb, ferritin, and TSAT from the baseline and were calculated using Wilcoxon signed rank test. Two-Way Repeated Measures ANOVA (Mixed-Factor ANOVA) was applied to compare between before and after effect in the two treatment arms (Table 3). 

Further, we looked and measured the changes from the baseline. Thus, this will overcome the different baseline level of lab data (Table 3).

- The conclusion that IS showed a higher increase in Hb is affected by the Hb baseline difference between the two groups

Authors’ response: Our conclusion is justified based on the statistical analysis and considerations taken in the analysis - % change/improvement and ANOVA method used. We hope these explanations have satisfied your concerns above.

7. PLOS authors have the option to publish the peer review history of their article (what does this mean?). If published, this will include your full peer review and any attached files.

Do you want your identity to be public for this peer review? For information about this choice, including consent withdrawal, please see our Privacy Policy.

Reviewer #2: No

Reviewer #3: No

The study was approved by HMC-IRB (i.e. MRC approval number 2101783; Protocol Number: MRC-01-19-171.

The patient records and all data used in our retrospective study were fully anonymised before we

transferred to SPSS program for data analysis and reporting.

---

## [Editor Report · Decision Letter 2]

12 Jul 2021

EFFICACY AND COST EFFECTIVENESS OF INTRAVENOUS FERRIC CARBOXYMALTOSE VERSUS IRON SUCROSE IN ADULT PATIENTS WITH IRON DEFICIENCY ANEMIA

PONE-D-20-35525R2

Dear Dr. BASHA,

We’re pleased to inform you that your manuscript has been judged scientifically suitable for publication and will be formally accepted for publication once it meets all outstanding technical requirements.

Kind regards,

Pavel Strnad

Academic Editor

PLOS ONE
---

## [Editor Report · Acceptance letter]

29 Jul 2021

PONE-D-20-35525R2 

EFFICACY AND COST EFFECTIVENESS OF INTRAVENOUS FERRIC CARBOXYMALTOSE VERSUS IRON SUCROSE IN ADULT PATIENTS WITH IRON DEFICIENCY ANAEMIA 

Dear Dr. Basha:

I'm pleased to inform you that your manuscript has been deemed suitable for publication in PLOS ONE. Congratulations! Your manuscript is now with our production department. 

Kind regards, 

on behalf of

Dr. Pavel Strnad 

Academic Editor

PLOS ONE